# Word and bit line operation of a 1 × 1 µm² superconducting vortex-based memory

Taras Golod [1], Lise Morlet-Decarnin [1] & Vladimir M. Krasnov [1] ✉

The lack of dense random access memory is one of the main bottlenecks for the creation of a digital superconducting computer. In this work we study experimentally vortex-based superconducting memory cells. Three main results are obtained. First, we test scalability and demonstrate that the cells can be straightforwardly miniaturized to submicron sizes. Second, we emphasize the importance of conscious geometrical engineering. In the studied devices we introduce an asymmetric easy track for vortex motion and show that it enables a controllable manipulation of vortex states. Finally, we perform a detailed analysis of word and bit line operation of a 1 × 1 $\mu$m² cell. High-endurance, non-volatile operation at zero magnetic field is reported. Remarkably, we observe that the combined word and bit line threshold current is significantly reduced compared to the bare word-line operation. This could greatly improve the selectivity of individual cell addressing in a multi-cell RAM. The achieved one square micron area is an important milestone and a significant step forward towards creation of a dense cryogenic memory.

The development of superconducting (SC) electronics could lead to a breakthrough in future computation techniques. Major advances in SC quantum computing were recently achieved[1,2]. However, practical calculations today are made on a classical computer and demands for digital computation capacities are growing in the explosive manner. Resistivity causes principle limitations for semiconductor electronics. The large resistance, $R$, of transistors and interconnections both limits the operation speed (determined by the $RC$ time constant) and creates a problem of heat management in very-large-scale-integration (VLSI) circuits. Those obstacles can be obviated by shifting to superconductors with $R = 0$. The maximum operation frequency of SC electronics is determined by the energy gap, which can exceed 10 THz in high-temperature SCs[3]. For large data facilities, shifting from semiconductors to superconductors could drastically improve both the power efficiency (by an order of magnitude) and the computation speed (by several orders of magnitude). Such perspectives reignited the interest to a classical SC computer[4–12].

Digital SC electronics has a long history. Rapid-single-flux-quantum (RSFQ) architecture was developed almost half a century ago[13]. It is based on storage and manipulation of the flux quantum, $\Phi_0$, in superconducting quantum interference devices (SQUIDs). Nb-based RSFQ electronics is capable of operation at sub-THz frequencies[4], two orders of magnitude faster than modern computers. However, RSFQ has a major problem with scalability. The current needed for the introduction of $\Phi_0$ is determined by the inductance, $\Lambda$, of the SQUID loop, $I = \Phi_0/\Lambda$. Upon miniaturization, $\Lambda$ decreases and the operation current increases inversely proportional to the size. Therefore, RSFQ is not compatible with the VLSI technology. The main bottleneck is the lack of dense random-access memory (RAM)[4–6]. State of the art RSFQ RAM has a footprint of ~100 $\mu$m² per bit[5,11,12,14]. Large sizes cause significant delay times. In fact, the speed of RSFQ is limited by delay times, rather than the energy gap[4].

Novel approaches are needed for building a VLSI-competitive SC electronics. Several strategies for making dense SC RAM were suggested[7,15–29] and are awaiting for the experimental scrutiny. In ref. 7 it was shown that a single Abrikosov vortex (AV) can be used as an information carrier. AV represents the smallest magnetic object in a superconductor, enabling miniaturization to submicron sizes. Prototypes of single-bit AVRAM cells with excellent performance were demonstrated[7]. Yet, the way from a single cell to a dense RAM is full of hazards, as could be learned from the history of MRAM[30]. Cross-talking between cells could hamper RAM operation. It could be mitigated by a highly selective word and bit line (WL + BL) addressing individual cells. So far WL + BL operation of AVRAM cells has not been analyzed.

¹Department of Physics, Stockholm University, AlbaNova University Center, SE-10691 Stockholm, Sweden. ✉e-mail: vladimir.krasnov@fysik.su.se

In this work, we study experimentally Nb-based AVRAM cells. Our aim is threefold. First, we study the scalability. Cells with the same geometry, but different sizes 5, 3, and 1 μm are tested. We confirm that AVRAM cells are scalable to submicron sizes, while keeping robust non-volatile operation at zero magnetic field. Second, we emphasize the importance of conscious geometrical design for the optimization of cells. In the studied cells, we introduce an asymmetric "easy track" for vortex motion. It removes the degeneracy between vortex and antivortex states and, thus, facilitates controllable operation of the cell. Finally, we analyze the word and bit line operation of a $1 \times 1\,\mu m^2$ cell. We observe that simultaneous application of WL and BL pulses can significantly reduce threshold currents for vortex manipulation. Such a cooperative WL + BL effect could greatly improve the selectivity of addressing individual cells in RAM. We conclude that AVRAM is a feasible candidate for the creation of dense cryogenic memory.

## Results

Planar AVRAM cells were made from a thin Nb film. The cells have a cross-like geometry with four electrodes, two variable-thickness-bridge-type Josephson junctions (JJs) and an artificial vortex trap, made by a Ga focused ion beam (FIB). More details about device fabrication and characterization can be found in refs. 7,31–33, "Methods" section and the Supplementary material. The fabrication procedure is highly reproducible and the characteristics of both JJs are practically identical. We can controllably introduce and remove vortices by short current pulses[7,31,32,34,35]. AV manipulation by light[36] and magnetic field[37] pulses is also possible.

### Miniaturization

To study the scalability, several cells with similar geometry but different lengths of readout JJs, $L_x = 5$, 3, and 1 μm, were made on the same chip. Figure 1a, b show scanning electron microscope (SEM) images of the cells with (a) $L_x = 3\,\mu m$ and (b) $L_x = 1\,\mu m$. Figure 1c, d show magnetic field dependencies of critical currents, $I_c(H)$, for these cells in the

vortex-free state, $V = 0$. All JJs exhibit Fraunhofer-type modulation with the central maximum at $H = 0$ and with the flux quantization field consistent with their geometry (see Supplementary sec. II). Larger cells, $L_x = 5\,\mu m$, are similar to those in ref. 7. Their characteristics can be found in the Supplementary sec. I.

Figure 1e, f show $I_c(H)$ for the 3 μm cell with a trapped vortex (e), or an antivortex (f). From Fig. 1d–f it is seen that the three primary states with different vorticities, (d) $V = 0$, (e) $V = 1$ and (f) $V = -1$, are clearly distinguishable. Vortex trapping leads to a characteristic distortion of $I_c(H)$[7,31,34,35,38], and to a threefold reduction of $I_c(H = 0)$. This enables a simple recognition and readout of vortex states. The closer the AV is to the JJ, the larger is the junction response[34]. In these cells, the vortex trap is placed symmetrically with respect to the JJs; see Fig. 1a. Consequently, the responses of both JJs are identical and $I_c(H)$ patterns for both JJs merge in Fig. 1e, f. Such a coincidence confirms that the vortex is indeed placed in the trap. We always use two JJs in our AVRAM cells: one for determining the state (the readout JJ) and the other for confirming that the vortex is located in the trap (the test JJ).

In Figure 2a, b, we show a SEM image and a sketch of a ~$1 \times 1\,\mu m^2$ cell, in which excessive parts of the Nb electrode were removed. The trap with a diameter of ~100 nm is placed close (~240 nm) to the top readout JJ. The bottom, test JJ is at a significantly longer distance (~740 nm). The left-right and the bottom-top electrode pares form WL and BL, respectively. Below we will focus on zero-field operation of this cell.

### Geometrical asymmetry

Although vortices can be manipulated by current pulses, in a symmetric cell at $H = 0$, 1, and −1 states are degenerate: a vortex and an antivortex will be generated simultaneously at opposite sides of the device, move inside and annihilate. Therefore, a perfectly symmetric cell would not be able to trap a vortex. In reality, there is always some asymmetry that lifts the degeneracy. However, the operation can not be fully controllable without conscious geometrical design.

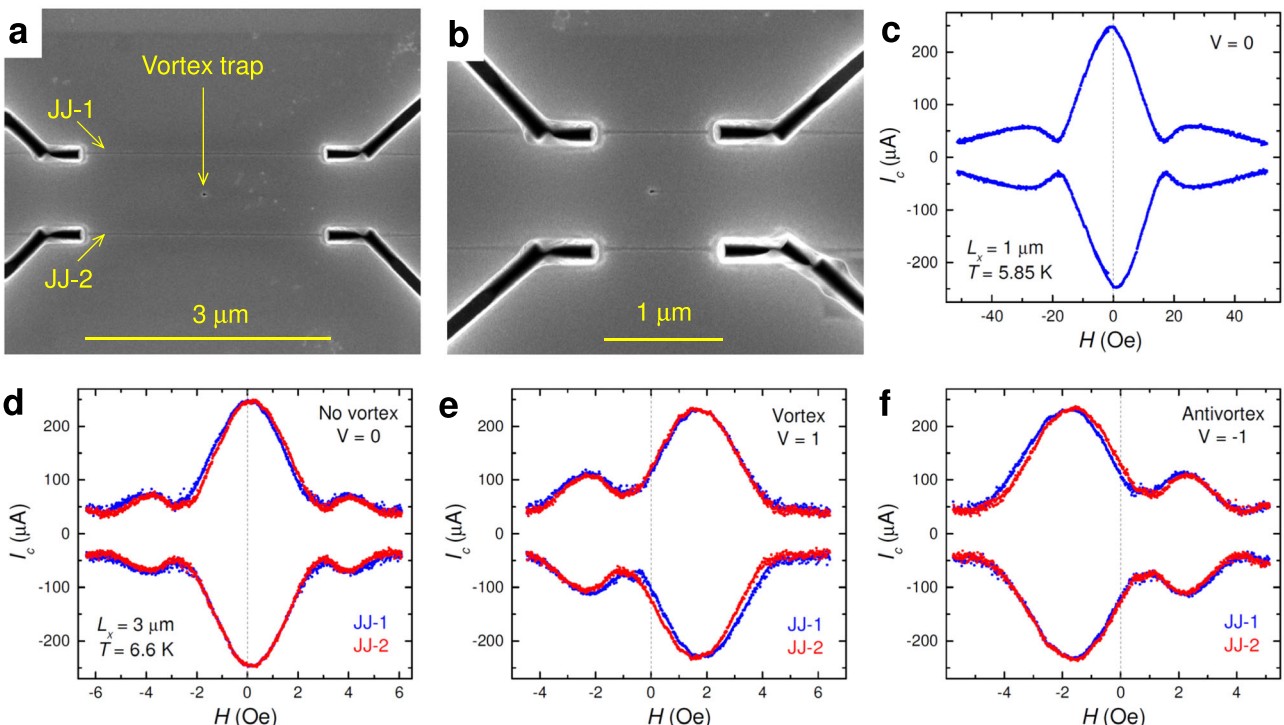

**Fig. 1 | Demonstration of AVRAM cell scalability. a, b** SEM images of two cells with sizes (**a**) $L_x \simeq 3\,\mu m$, and (**b**) $L_x \simeq 1\,\mu m$. **c** Magnetic field modulation of the critical current in a readout junction on the $L_x = 1\,\mu m$ cell in the vortex free state, $V = 0$. **d–f** $I_c(H)$ modulations for the $L_x = 3\,\mu m$ cell: (**d**) in the vortex-free state, $V = 0$, (**e**) with a trapped vortex, $V = 1$, and (**f**) with an antivortex $V = -1$. Blue and red symbols in **d–f** show simultaneously measured $I_c$ for both readout JJs.

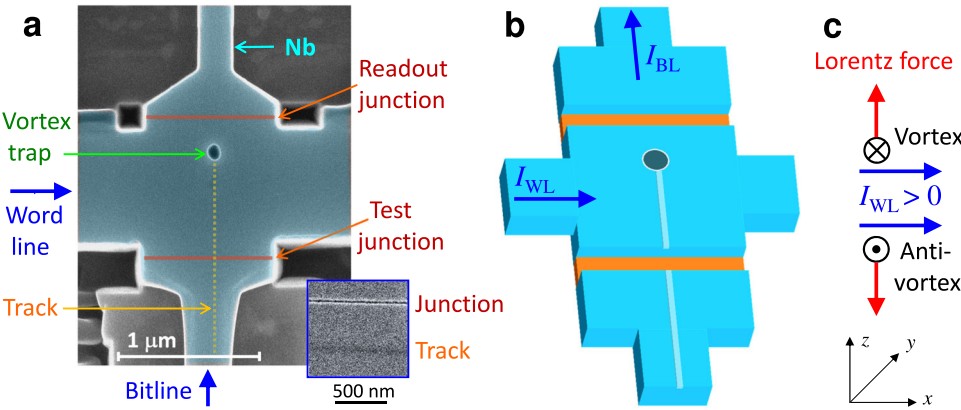

**Fig. 2 | Geometry of a $1 \times 1\,\mu m^2$ cell with word and bit lines. a** SEM image of the device (false color). Inset shows a SEM image of a junction groove and a FIB track (for another device). **b** A sketch of the cell with indicated directions of wordline and bitline currents. **c** Direction of Lorentz forces, exerted on a vortex and an antivortex by a positive WL current. The easy operation with a small WL threshold current is expected when the vortex is transported along the track.

The asymmetry in the device from Fig. 2a is introduced by making an easy track for vortex motion at one side of the trap. It is made by single swiping of the FIB to/from the trap. The inset in Fig. 2a shows a high-resolution SEM image of the track alongside the junction groove (for another device). It is seen that the track makes just a minor depression in the Nb film. It does not form a JJ, but creates an easy path for AV to and from the trap[31]. Figure 2c indicates directions of Lorentz forces, $F_L$, exerted on a vortex and an antivortex by a positive WL current. From the comparison with the track position in Fig. 2a, it is expected that a positive $I_{WL}$ would easily introduce a vortex $(0 \rightarrow 1)$ and remove an antivortex $(-1 \rightarrow 0)$ along the easy track. All other operations should be more difficult and require larger threshold currents.

Table 1 summarizes measured threshold currents for all operations on this cell, performed by sending a single pulse along either WL or BL. For WL operation, we observe three threshold levels: $|I_{WL}| \simeq 0.15\,mA$ (easy), $\simeq 0.73\,mA$ (moderate), and $\simeq 0.87\,mA$ (difficult). Of those, the "easy" corresponds to erasing, $\pm 1 \rightarrow 0$, along the track; the "moderate" to writing $0 \rightarrow \pm 1$ along the track; and the "difficult" to all other operations in the direction opposite to the track. Thus, the subtle track is playing a very important role. It introduces the required asymmetry, enabling a controllable realization of all four main operations along the easy path. Control in the "difficult" regime is much worse because such currents can do any operation.

Lorentz forces induced by BL currents are perpendicular to the track. Therefore, there is no easy path and all BL operations are "hard", $|I_{BL}| > 1\,mA$. This is important because the passiveness of BL enables nondestructive readout. As seen from Fig. 2a, b, the BL current goes simultaneously through both JJs. Since fairly large BL currents do not affect the vortex state, we can nondestructively readout the state by measuring junction voltages, or resistances. This is demonstrated in

Figure 3a, which shows the current–voltage ($I$–$V$) characteristics of the readout JJ in $V = 0$ (black), 1 (red), and $-1$ (blue) states. The current, sent via the BL, does not cause switching between vortex states within this bias range. Thus, the introduced asymmetry is crucial both for controllable operation and for nondestructive readout.

### Wordline operation

First, we analyze the $0 \rightarrow \pm 1$ write operation solely by the WL current. The experiment is done in the following manner. The cell is prepared in the 0-state, and after that a short pulse with an amplitude $I_{WL}$ is sent through the WL. The state of the device is evaluated by measuring the locking resistance, $R$, of the JJ via the BL (see Methods). The probe ac current, $I_{ac} \simeq 130\,\mu A$, is smaller than $I_c$ in the 0-state, but larger than in 1 and $-1$ states (see Fig. 3a), so that the 0-state corresponds to $R = 0\,\Omega$, and 1 and $-1$ states to $R \simeq 0.27$ and $0.26\,\Omega$, respectively. Figure 3b shows the readout JJ resistance as a function of $I_{WL}$. It is seen that vortices can be written in the range $0.73\,mA \lesssim |I_{WL}| \lesssim 0.87\,mA$. The lower limit, $|I_{WL}| \simeq 0.73\,mA$, corresponds to the "moderate" threshold for writing along the track. As follows from Table 1, positive $I_{WL}$ causes $0 \rightarrow 1$ switching, and negative, $0 \rightarrow -1$ switching. At $|I_{WL}| > 0.87\,mA$, no switching is observed. Presumably, this is caused by exceeding the "difficult" threshold, which leads to the introduction of an opposite vortex, that annihilates with the one trapped earlier via the track. A certain stratification of $I_{WL}$ can be attributed to the stroboscopic effect due to the large duration of the pulse, compared to the vortex time-of-flight, as discussed in the Supplementary sec. VII.

In Fig. 4, we show representative examples of high-endurance operations by periodic positive/negative WL pulses. In Fig. 4a, a positive pulse, $I_{WL} = 0.75\,mA$, slightly above the "moderate" threshold, writes a vortex $(0 \rightarrow 1)$ and a small negative pulse, $I_{WL} = 187.5\,\mu A$, erases it. Both operations occur along the track. It is seen that erasing is significantly easier than writing.

In Fig. 4b, a negative pulse, $I_{WL} = -0.8\,mA$, writes an antivortex $(0 \rightarrow -1)$. The subsequent positive pulse, $I_{WL} = 1.0\,mA$, above the "difficult" threshold, annihilates it and switches the sell into the 0-state. The next similar pulse doesn't cause switching from the 0-state, consistent with Fig. 3b.

In Fig. 4c, a 4-pulse train with small $\pm 0.2\,mA$ and moderate $\pm 0.8\,mA$ pulses is applied. Here, the moderate negative/positive pulses write $-1/1$, and subsequent small positive/negative pulses erase them. All operations are achieved via the easy track.

Figure 4d represents a similar 4-pulse train with $\pm 0.15\,mA$ and $\pm 0.8\,mA$ pulses. Here the small negative pulse appears to be sub-threshold and does not erase the 1-state. The subsequent moderate negative pulse causes $1 \rightarrow -1$ switching, which can be considered as a

## Table 1 | Summary of single-line operations for the device from Fig. 2a at $T = 5.85\,K$ and $H = 0$

| Operation | Easy | Moderate | Difficult | Hard |
|---|---|---|---|---|
| Write 1 $(0 \rightarrow 1)$ | – | $I_{WL} > 0$ | $I_{WL} < 0$ | BL |
| Write $-1$ $(0 \rightarrow -1)$ | – | $I_{WL} < 0$ | $I_{WL} > 0$ | BL |
| Erase 1 $(1 \rightarrow 0)$ | $I_{WL} < 0$ | – | $I_{WL} > 0$ | BL |
| Erase $-1$ $(-1 \rightarrow 0)$ | $I_{WL} > 0$ | – | $I_{WL} < 0$ | BL |
| Threshold (mA) | 0.15 | 0.73 | 0.87 | >1 |

WL operations have three distinct levels. Easy ($|I_{WL}| \simeq 0.15\,mA$) and moderate ($|I_{WL}| \simeq 0.73\,mA$) thresholds correspond to vortex propagation along the easy track. The difficult threshold ($|I_{WL}| \simeq 0.87\,mA$) - to vortex motion outside the track. For BL operations there is no easy path; therefore, all BL operations are hard ($|I_{BL}| > 1\,mA$).

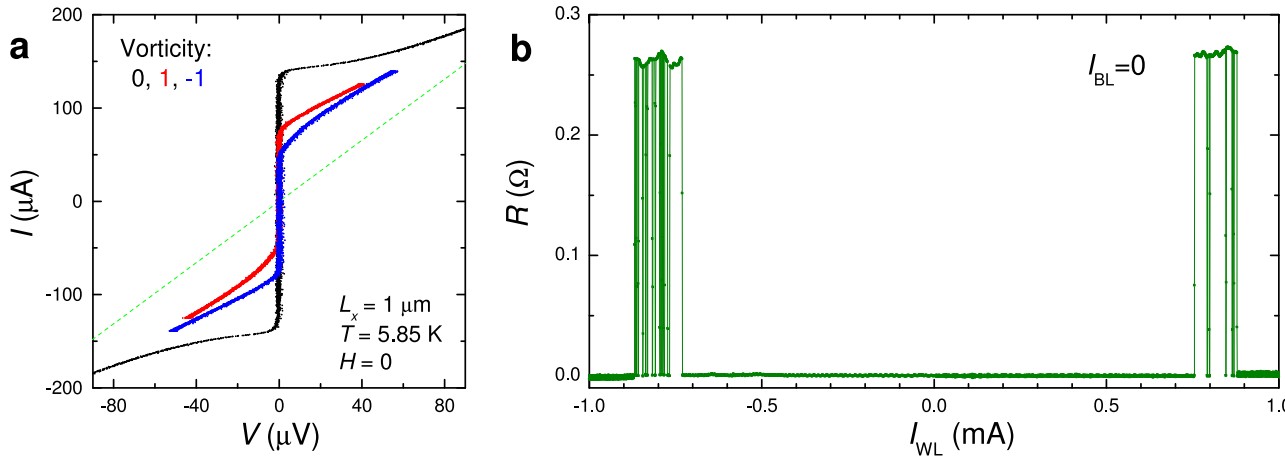

**Fig. 3 | Characteristics of the 1 × 1 μm² cell from Fig. 2a. a** Current–Voltage characteristics of the readout junction: black - in the vortex-free state, red - with the trapped vortex, and blue - with an antivortex. Green dashed line represents the normal resistance, $R_n = 0.61\,\Omega$. **b** Demonstration of the write operation $0 \rightarrow \pm 1$ by wordline pulses (without bitline currents). The readout junction resistance is shown as a function of the WL current pulse amplitude. Measurements were performed at $T = 5.85\,K$ and $H = 0$.

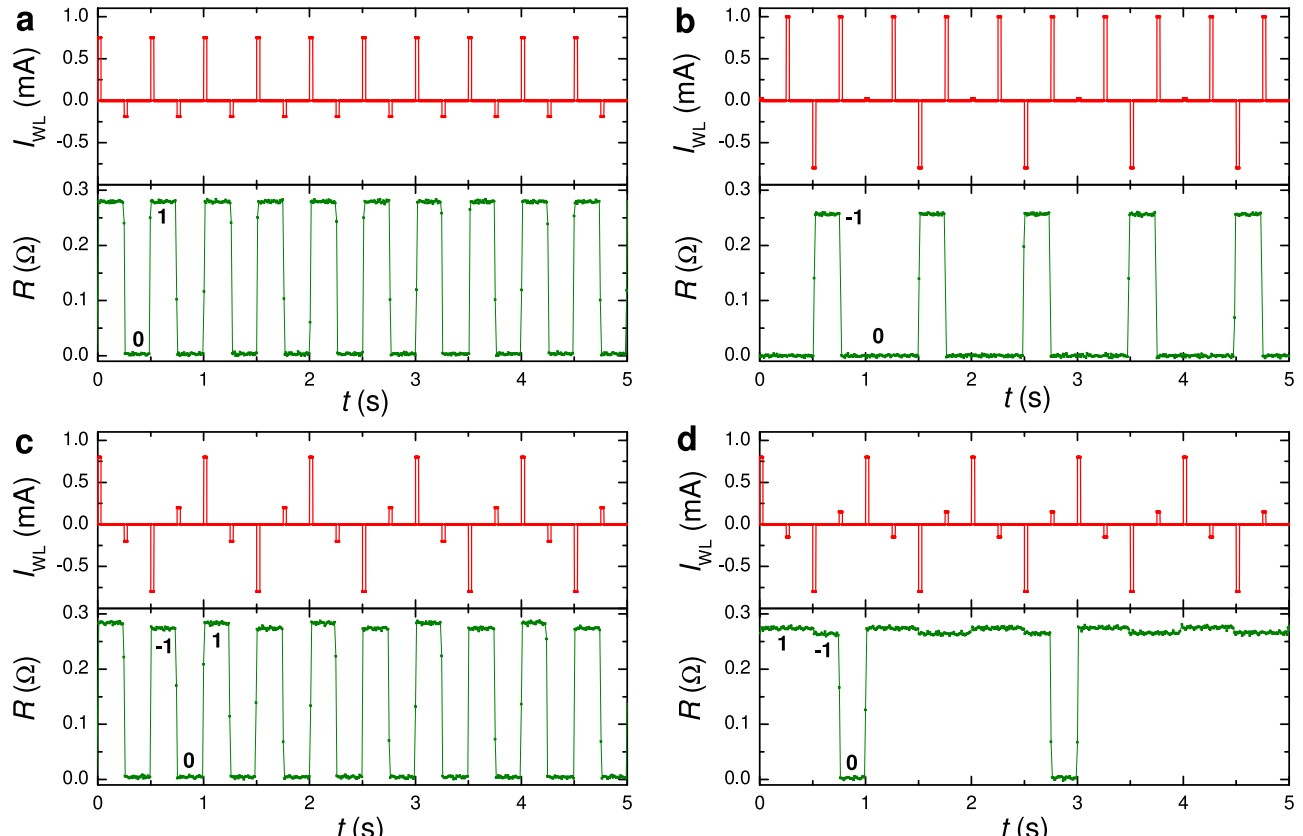

**Fig. 4 | Wordline operation of the 1 × 1 μm² cell at zero bitline current. a** Writing and erasing a vortex along the easy path by WL pulses 0.75 and −0.1875 mA. **b** Writing an antivortex along the easy path and erasing along the difficult path by WL pulses −0.8 and 1.0 mA. **c** Writing and erasing vortices and antivortices along the easy path by WL pulses ± 0.8 and ± 0.2 mA. **d** The same as in **c** with WL pulses ± 0.8 and ± 0.15 mA. In all figures, top panels show time sequences of WL pulses and bottom panels - a simultaneously measured ac-resistance of the readout junction. Measurements were performed at $T = 5.85\,K$ and $H = 0$.

sequential entrance of two antivortices, of which the first annihilates with the trapped vortex and the second stays in the trap. The small positive pulse, however, is sufficient for erasing the −1 state. The distinction between 1 and −1 states, seen from resistances in the bottom panel of Fig. 4d as well as from the $I–V$s in Fig. 3a, indicates the presence of an additional left-right asymmetry in the cell, leading to a current-induced self-field effect, which removes the degeneracy

between 1 and −1 states at $H = 0$[32] (as discussed in the Supplementary sec. V).

## Word and bit-line operation

RAM requires selective addressing of individual cells by coincident WL and BL pulses. Figure 5 demonstrates the WL + BL write operation, $0 \rightarrow \pm 1$. JJ resistance is shown as a function of the positive WL amplitude

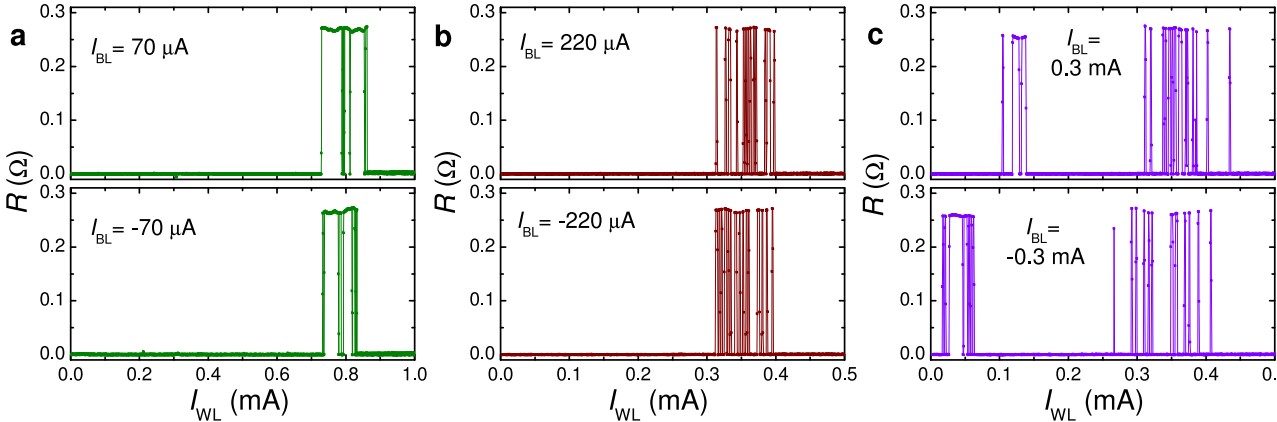

**Fig. 5 | Word and bit-line operation of the 1 × 1 µm² cell.** The readout junction resistance is shown as a function of the positive WL pulse amplitude, $I_{WL}$, for fixed amplitudes of the BL current. (**a**) $I_{BL} = \pm 70$ µA; (**b**) $I_{BL} = \pm 220$ µA; and (**c**) $I_{BL} = \pm 300$ µA. A significant reduction of the threshold WL current is observed upon simultaneous application of the BL current. Measurements were performed at $T = 5.85$ K and $H = 0$.

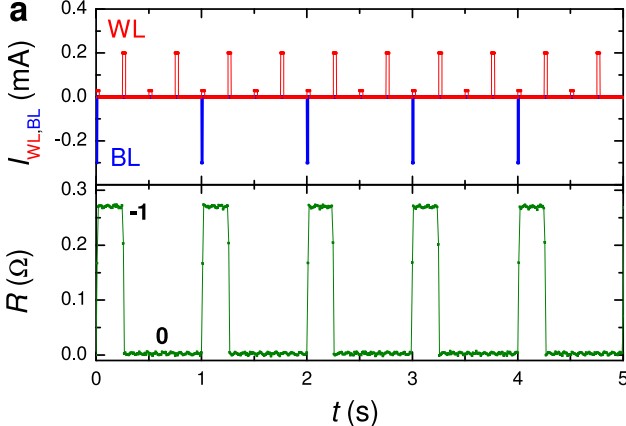
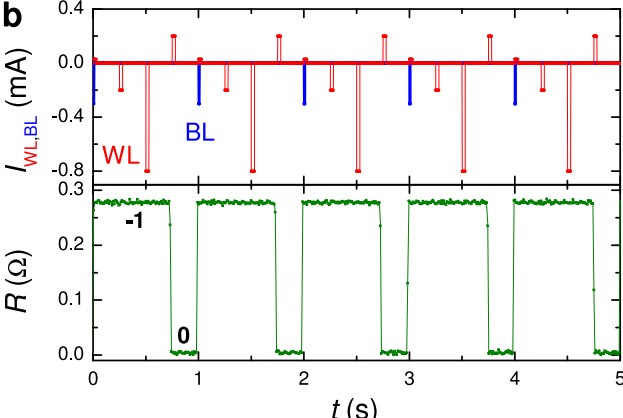

**Fig. 6 | Demonstration of a simultaneous word and bit-line operation of the 1 × 1 µm² cell. a** Writing an antivortex by combined pulses $I_{BL} = -300$ µA and $I_{WL} = 29$ µA; and erasing by a WL pulse 200 µA. **b** Similar write and erase pulses as in **a** with additional intermediate WL pulses. It is seen that the negative WL pulses do not change the −1 state because they tend to write an antivortex and the trap is already occupied. In all figures, top panels show time sequences of WL (red) and BL (blue) pulses. Bottom panels show a simultaneously measured ac-resistance of the readout junction. Measurements were performed at $T = 5.85$ K and $H = 0$.

for several values of BL pulses: (a) $I_{BL} = \pm 70$ µA, (b) $I_{BL} = \pm 220$ µA, and (c) $I_{BL} = \pm 300$ µA. It is seen that the threshold WL current is reducing with increasing $I_{BL}$. Remarkably, the WL + BL operation leads to a significant reduction of the total threshold, $I_{tot} = |I_{WL}| + |I_{BL}|$: $I_{tot} \simeq 730$ µA for $I_{BL} = 0$ (see Fig. 3) and $\pm 70$ µA; $I_{tot} \simeq 520$ µA for $I_{BL} = \pm 220$ µA; and $I_{tot} \simeq 320 - 400$ µA for $I_{BL} = \pm 300$ µA. The latter is approximately two times smaller than for a solo WL operation. Thus, the cooperative effect of WL and BL pulses is significantly better than a sum of the two currents. This is very good because it can greatly improve the selectivity of addressing individual cells in RAM. As discussed in the Supplementary sec. V, the cooperative WL+BL effect is caused by the opening of an additional channel for vortex entrance from the top readout JJ.

Figure 6 demonstrates switching dynamics with WL and BL pulses. Panel (a) shows low-threshold switching. Here a small $I_{BL} = -300$ µA together with a very small $I_{WL} = 29$ µA causes $0 \rightarrow -1$ switching and a small $I_{WL} = 200$ µA erases it, $-1 \rightarrow 0$. Subsequent WL-only pulses with the same amplitudes, 29 and 200 µA, do not affect the 0-state (because there is nothing to erase). In panel (b), similar write and erase pulses are applied. Negative WL-only pulses −200 and −800 µA, applied in the −1 state, do not affect the state. Even larger BL-only pulses, $|I_{BL}| > 1$ mA, can be applied without changing the state of the device (see Table 1). This illustrates that the smallness of the combined WL + BL current does not affect the robustness of nondestructive readout.

## Discussion
### Scalability
We demonstrated that planar AVRAM cells can be scaled down to ~1 × 1 µm². Threshold currents of small cells are significantly reduced, compared to larger cells, studied in ref. 7. We observe that the threshold $I_{WL}$ is approximately proportional to the width of the WL electrode. This is expected because vortex motion occurs upon exceeding of the depinning current density, which is a material property. Consequently, threshold currents scale down upon miniaturization and do not suffer from the current divergency problem, as SQUIDs. This proves that AVRAM cells are indeed scalable.

The area of the presented ~1 × 1 µm² cell is already ~100 times smaller than for the state of the art RSFQ cells[11,12]. But this is not the ultimate limit. Miniaturization of AVRAM cells is limited by the stability of AV in a mesoscopic SC electrode: the vortex should be persistent in the trap at zero magnetic field. Metastable vortex configurations in mesoscopic superconductors have been intensively studied and are well understood[39-45]. The trap plays an important role in stabilizing the vortex. It changes the electrode topology and creates a sharp pinning potential, which stabilizes otherwise energetically unfavorable ±1 vortex states at zero field. The tendency of the vortex to spontaneously jump out of the electrode can be viewed as the consequence of an

attractive interaction with an image antivortex outside the electrode[34]. From Fig. 2a it can be seen that for the studied $1 \times 1\,\mu m^2$ device the nearest to the trap edge corresponds to the readout JJ, 240 nm away. Apparently, this distance is sufficient for preventing a spontaneous vortex exit. Consequently, the cell can be straightforwardly miniaturized to twice this size, i.e., to ~$500 \times 500$ nm$^2$.

Generally, vortices can exist in superconductors with sizes down to the coherence length, $\xi$. However, the smaller is the size, the larger is the relative energy cost of a vortex at zero field. Our sputtered Nb films have a short $\xi(0) = 14$ nm and a London penetration depth, $\lambda_L(0) \simeq 100$ nm[46]. Therefore, we anticipate that further miniaturization down to ~100 nm should be feasible. Vortex stray fields will, presumably, not play a significant role for AVRAM density because vortices loose the magnetic flux in a deeply mesoscopic limit, as discussed in the Supplementary sec. IX.

## Conscious geometrical design

We have emphasized that a specific geometrical asymmetry is needed for controllable vortex manipulation. This is particularly important for submicron AVRAM cells, which are in the mesoscopic limit. Geometry (sizes and shapes) is playing a decisive role for such fluxonic quantum dots[39–45]. Proper geometrical design would be needed for optimization of AVRAM cells.

In this work, we introduced several small but important improvements in the cell design, compared to the initial prototype[7]. First, the film structure was reduced to a single Nb film (compared to a bilayer superconductor/normal metal or superconductor/ferromagnet) and the readout junction structure became a variable-thickness-bridge type (compared to proximity-coupled SNS or SFS JJs). Apart from a bare simplification, this leads to a major enhancement of the $I_c R_n$ product, reaching almost 1 mV at low $T$[33]. The corresponding enhancement of the readout voltage[32] is crucial for device operation.

Most importantly, we have consciously engineered geometrical asymmetry. An easy track for vortex manipulation was made at one side of the trap. This led to improved controllability and reduced threshold currents. The analysis in Table 1 confirms that the track works and vortices indeed follow this route. The achieved threshold currents, as low as 150 μA, are optimal for practical devices. Further reduction would jeopardize the nondestructive readout and lead to smaller readout signals. Other geometrical features that help to guide vortex motion were also suggested in the literature[44,45] and could be implemented, if necessary.

To conclude, we have studied single-bit AVRAM cells and arrived to three main conclusions.

(i) The planar AVRAM cell design is straightforwardly scalable to submicron sizes. The threshold current is reduced upon miniaturization, approximately proportionally to the size, thus obviating the problem of SQUID-based RSFQ cells. Robust, non-volatile operation at zero magnetic field was demonstrated for a ~$1 \times 1\,\mu m^2$ cell. Its area is approximately 100 times smaller than for the state of the art RSFQ cells.

(ii) We emphasized the importance of a conscious geometrical design of the cells. A specific geometrical asymmetry (an easy track) was introduced and enabled controllable and comprehensible operation of the cell.

(iii) Word and bit line operation exhibits a profound cooperative effect, reducing the total threshold current. The smaller combined WL+BL currents reduce chances of affecting other cells at the same WL. This can improve the selectivity of cell addressing in a multi-cell RAM.

We conclude that planar AVRAM cells are promising candidates for the creation of VLSI-compatible superconducting memory. The achieved one square micron area is an important milestone and a significant step forward in this direction.

## Methods

### Sample fabrication

The studied devices were made from a single Nb film (70 nm). It was deposited by the dc-magnetron sputtering at room temperature on an oxidized Si wafer. First, the film was patterned into ~5 μm-wide bridges by photolithography and reactive ion etching. Subsequently, it was nano-patterned by Ga FIB. The readout junctions have a variable-thickness-bridge structure. They were made by cutting narrow grooves in the Nb film by FIB. The cuts are made as single-pixel lines with the nominal etching depth of typically 60 nm, using standard Si-milling parameters (volume-per-dose 0.27 μm$^3$/nC) at an ion current of 10 pA (30 keV). The etching time is calculated automatically for the chosen depth and volume-per-dose. The actual depth of the cut is, however, reduced by redeposition of Nb. The groove has a V-shape profile with the aspect ratio (depth/width) ~2. From the inset in Fig. 2a, it can be seen that the groove width at the surface is (~20−30 nm), from which the actual depth can be estimated ~40−60 nm. Self-limiting of the cut depth is consistent with a slow increase of junction resistances with increasing the nominal depth. In the cell from Fig. 2a, the area of Nb electrodes was reduced by additional FIB milling. During this process, the whole cell, including junctions, was etched by several FIB scans needed for pattern alignment. This resulted in a slightly reduced $I_c$, compared to the device with the same $L_x$ from Fig. 1b.

### Transport characterization

Measurements are performed in a closed-cycle cryostat. The four-terminal cross-like configuration allows both word and bit line operation and an independent measurement of the characteristics of both JJs. For this the transport current is sent simultaneously through both JJs via the bit-line (from top to bottom electrode) and junction voltages are measured in a quasi four-probe manner via one BL and one WL electrode. Measurements of $I_c(H)$ patterns are performed in a magnetic field perpendicular to the Nb film, supplied by a superconducting magnet.

Current pulses for WL and BL operation are created by two programmable current sources. The BL current contains short pulses superimposed on a slowly varying ac-current with $f = 23$, or 113 Hz, which allows continuous monitoring of the read-out JJ resistance. The probe ac-current does not cause or affect switching of the cells because it has a small, sub-threshold amplitude and the pulses are applied at zero's of the probe current, as shown in the Supplementary Fig. 4.

## Data availability

The data that support the findings of this study are available from the corresponding author upon reasonable request.

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

## Acknowledgements
The work was supported by the Swedish Research Council. Open access funding provided by Stockholm University.

## Author contributions
T.G. fabricated samples and performed measurements together with L.M.-D. V.M.K. conceived the project and wrote the paper with input from T.G. and L.M.-D.

## Funding

## Competing interests
The authors declare no competing interests.
