## [Peer Review File · Nature Communications]

REVIEWER COMMENTS

Reviewer #1 (Remarks to the Author):

The authors present a memory cell based on storing the Abrikosov (anti)Vortex in a specially prepared artificial pinning site (hole in the superconducting film) and sensing its state (vortex, 0, anti-vortex) by a Josephson junction situated in its vicinity.

The main results of this work is the memory cell, optimized in many respects (size, asymmetric vortex-junction coupling, vortex guiding to/from pinning site) in comparison with the previous version of the same authors. It shows reliable and reproducible operation, see Figs. 4 and 6. However, the domain of operation e.g. in terms of the values of I_{WL} , consists of multiple small sub-domains (Figs. 3 and 5), i.e. the circuit has very small margins, which rises questions about its real application. Later probably requires clarification and improvements in the future.

In spite of this, I believe that this works represents a significant step forward towards developing of superconducting memory based on Abrikosov vortices and in general for any devices that exploit Abrikosov vortices to project the phase into Josephson junction and, thus, change its properties.

The paper is very well written, the conclusions and claims are well supported by presented experimental data.

I have found some minor issues that need to be clarified.

1. While discussing $I_c(H)$ in section 2b, it is not clear how the current was applied, through both junctions simultaneously or through each of them separately (using the middle electrode)?
2. At the end of the same paragraph, how the "flux quantization field" is defined and why it is not proportional to the area? Or how (effective) area is defined?
3. In Fig.2 the track and the junction are perpendicular. However in the inset of Fig.2 they are parallel. Huh?

4. In Sec.IId it is not clear what is the typical frequency of the probe ac signal and whether it is applied permanently or only for readout when $I_{WL}=0$.

5. It is not clear why, in Sec. IIe, "the cooperative effect of WL and BL pulses is significantly better than a sum of the two currents"? Is here some time/frequency/phase interplay takes place?

Reviewer #2 (Remarks to the Author):

This work demonstrates a $1\mu\text{m}^2$ superconducting vortex-based memory cell. They show the miniaturization of the superconducting memory for a dense cryogenic memory. My comments on this work are as follows-

- The authors mention in the introduction that the large footprint of the RSFQ RAMs is a major issue, but they ignore other pressing problems like flux trapping and inductive coupling. These issues should also be addressed. For superconducting memories, flux trapping limits the integration density since a minimum space should be maintained between two devices. Will there be any such scenario in this case?

- I believe the characteristics shown in Fig. 1(c) and Fig. 3(a) correspond to the same device and operating condition. However, there is a discrepancy in the critical current levels. At $H=0$, the critical current is way above $200\ \mu\text{A}$ in Fig. 1(c) but in Fig. 3(a), it is less than $150\ \mu\text{A}$. Please clarify.

- The authors discuss the WL and BL operation of a memory cell. But they should discuss the WL and BL operation in an array scenario. Will the array require any access device to successfully write/read any specific cell in the array? If yes, will using an access device affect the integration density in the array scenario?

- It is not clear how the write/read operation of a specific cell in an array can be done without disturbing other cells. How will the WL and BL pulses be applied in write and read operations? What will be the sensing mechanism?

- The authors only show the write operation of the memory cell. They should also show the read operation.

- The manuscript does not provide enough discussion on the existing cryogenic memory technologies and their limitations.

Reviewer #3 (Remarks to the Author):

The authors experimentally demonstrated the operation of submicron superconducting memory cells based on an Abrikosov vortex (AV) trapped in the superconducting film. They placed planer-type Josephson junctions close to a trap center and observed the change of a Josephson threshold current with and without AV in the trap center. Three significant results were shown in the article. Firstly, they showed the stable write/read operation in the memory cell of 1 μm size. Second, they demonstrated controllable write/read operation in the memory cell by introducing the asymmetrical cell structure. Finally, they showed the selectivity of the memory cell through bit-line and word-line, where applying the bit-line current to the cell considerably reduced the word-line current for data writing. These features are promising for fabricating high-dense superconducting memories that are not realized yet. However, several questions and comments shown below about the proposed system and experimental results have to be clarified.

1. The reason for the large size of the RSFQ circuit is due to the inductance formed on the ground plane. This is effective for reducing magnetic coupling between memory cells. In a memory using vortices on a plane, it is considered that the effect of magnetic coupling between memory cells affects the memory operation as the memory scale increases. The authors need to comment on how to solve this.

2. The R versus I_{WL} characteristic in Fig. 3(b) becomes oscillatory when I_{WL} exceeds the threshold. Also, the period of the oscillations is irregular. What causes this oscillation?

3. Why does applying bit-line current I_{BL} reduce the total threshold current I_{tot} in the word and bit-line operation? A qualitative explanation is necessary.

4. When the bit-line current I_{BL} was zero, the word-line currents required for the writing “-1” operation and the data-erasing operation were $I_{WL} = -730\mu\text{A}$ and $150\mu\text{A}$, respectively, as shown in Table 1. On the other hand, when $I_{BL} = -300\mu\text{A}$, $I_{WL} = 29\mu\text{A}$ and $200\mu\text{A}$, respectively. Why did the application of I_{BL} reverse the sign of the word-line current for the writing “-1” operation? Also, why did the current for the data-erasing operation increase the word-line current? A qualitative explanation is necessary.

5. Why are the magnitudes of the threshold word-line current different for the positive and negative bit-line current I_{BL} in Fig. 5(c)?

6. In Fig. 6 (b), if the authors examine the robustness of the memory operation, the threshold for the data-erasing operation concerning the application of positive I_{WL} should be examined.

Point-to-point reply to Reviewers.

We want to thank all three Reviewers for the constructive critics and valuable remarks that helped to improve the manuscript. We introduced corresponding changes and clarifications in the modified version. In order not to overload the main manuscript, most of the clarifications are provided in the Supplementary with only short notes added in the main text. Those modifications are marked in red (both in the main text and the Supplementary).

Reviewer #1

Reviewer #1 writes:

“The authors present a memory cell based on storing the Abrikosov (anti)Vortex in a specially prepared artificial pinning site (hole in the superconducting film) and sensing its state (vortex, 0, anti-vortex) by a Josephson junction situated in its vicinity.

The main results of this work is the memory cell, optimized in many respects (size, asymmetric vortex-junction coupling, vortex guiding to/from pinning site) in comparison with the previous version of the same authors. It shows reliable and reproducible operation, see Figs. 4 and 6. However, the domain of operation e.g. in terms of the values of I_{WL} , consists of multiple small sub-domains (Figs. 3 and 5), i.e. the circuit has very small margins, which raises questions about its real application. Later probably requires clarification and improvements in the future.

In spite of this, I believe that this works represents a significant step forward towards developing of superconducting memory based on Abrikosov vortices and in general for any devices that exploit Abrikosov vortices to project the phase into Josephson junction and, thus, change its properties.

The paper is very well written, the conclusions and claims are well supported by presented experimental data.”

Reply-1:

We agree that the issue with stratification of the switching current is important. It appears at certain conditions and can indeed reduce the tolerance margins. The likely origin of this effect is related to a very long duration of applied current pulses (~100 us) compared to the vortex time of flight in the device (sub-ns). At high current amplitudes the device goes in the flux-flow state with many vortices passing through the device. This is clear from the $1 \rightarrow -1$ switching, shown in Fig. 4 (d), which implies that at least two antivortices were involved: the first annihilated the trapped vortex and the subsequent get trapped. The flux-flow state can lead to formation of the phase-slip line – a hot track along which vortices and antivortices periodically enter from opposite sides of the device and kick out or annihilate an already trapped vortex. In the flux-flow case, the observed stratification could be the stroboscopic effect determined by the ratio of the pulse length to the vortex time of flight. If so, this will not play a role when the vortex is manipulated by short pulses, which is the ultimate goal for such devices. Usually the stratification can be removed by a modest variation of experimental conditions. We demonstrate this in the new Supplementary Figure 7, which shows that the stratification can be completely removed by either changing the pulse width, or the time-of-flight. The latter can be achieved by a modest heating of the device, which slightly increases the vortex viscosity. This is in line with expectations for the stroboscopic effect. We have added this clarification in the modified version of the Supplementary (sec. VII).

Reviewer #1 writes:

“I have found some minor issues that need to be clarified.

1. While discussing $I_c(H)$ in section 2b, it is not clear how the current was applied, through both junctions simultaneously or through each of them separately (using the middle electrode)?”

Reply-2:

In all cases the probe current is applied via the bit-line (bottom-top electrodes), simultaneously through both JJs. In this case characteristics of both junctions can be measured simultaneously (but independently) using additional WL electrodes. A clarification is added in the modified version (main text and Methods).

Reviewer #1 writes:

“2. At the end of the same paragraph, how the "flux quantization field" is defined and why it is not proportional to the area? Or how (effective) area is defined?”

Reply-3:

This is the specifics of planar geometry. The flux quantization field agrees with the anticipated flux quantization area, described by Eq.(S1) of the Supplementary. To make it more visible, we made a special subheading (new sec. II).

Reviewer #1 writes:

“3. In Fig.2 the track and the junction are perpendicular. However in the inset of Fig.2 they are parallel. Huh?”

Reply-4:

The inset shows another device (as noted in the captions), in which the junction and the track are close to each other and parallel. This allows demonstration with a high-resolution. When the two lines are perpendicular, it is difficult to show them simultaneously with a high magnification.

Reviewer #1 writes:

“4. In Sec.IId it is not clear what is the typical frequency of the probe ac signal and whether it is applied permanently or only for readout when $I_{WL}=0$.”

Reply-5:

The typical frequency is 23 or 113 Hz, applied continuously. This is described in the Supplementary, see the Suppl. Fig. 4. We added a short clarification in the text (Methods).

Reviewer #1 writes:

“5. It is not clear why, in Sec. IIe, "the cooperative effect of WL and BL pulses is significantly better than a sum of the two currents"? Is here some time/frequency/phase interplay takes place?”

Reply-6:

As discussed in Replies 4 and 5 to Reviewer-3, the cooperative effect can be explained by the opening on a new easy path for AV from the top readout JJ. The clarification is provided in the new sec. of the Supplementary: “V. The origin of cooperative word and bit-line effect”.

Reviewer #2

Reviewer #2 writes:

“This work demonstrates a $1\mu\text{m}^2$ superconducting vortex-based memory cell. They show the miniaturization of the superconducting memory for a dense cryogenic memory. My comments on this work are as follows-

- The authors mention in the introduction that the large footprint of the RSFQ RAMs is a major issue, but they ignore other pressing problems like flux trapping and inductive coupling. These issues should also be addressed. For superconducting memories, flux trapping limits the integration density since a minimum space should be maintained between two devices. Will there be any such scenario in this case?”

Reply -1:

This is an interesting point. Our problem is, in fact, the opposite: we are struggling to trap the vortex. The difference is caused by different sizes, compared to the London penetration depth. RSFQ components are large, $\gg \lambda$, and, therefore, prone to spontaneous vortex trapping. To the contrary, we are dealing with small mesoscopic electrodes $\sim \lambda$, in which spontaneous flux trapping does not occur. The magnetic field for introducing a flux quantum in a $1 \times 1\mu\text{m}^2$ island is quite large, 20 Oe, and the vortex will escape when the field is removed. In our devices the vortex stays only because we made a trap. As discussed in the end of the manuscript, the limit for miniaturization of vortex-based devices is set by the growing difficulty of vortex trapping in smaller islands. We added a short clarification in the modified version of the manuscript.

Reviewer #2 writes:

“- I believe the characteristics shown in Fig. 1(c) and Fig. 3(a) correspond to the same device and operating condition. However, there is a discrepancy in the critical current levels. At $H=0$, the critical current is way above 200 μA in Fig. 1(c) but in Fig. 3(a), it is less than 150 μA . Please clarify.”

Reply-2:

These are two different devices: Fig. 1(c) shows data for the cell from Fig. 1 (b) and Fig. 3(a) for the one from Fig. 2(a). Technically, the difference is caused by a slightly smaller junction length and a longer processing time in FIB, accompanied by an additional ion milling, for the cell from Fig. 2(a).

Reviewer #2 writes:

“- The authors discuss the WL and BL operation of a memory cell. But they should discuss the WL and BL operation in an array scenario. Will the array require any access device to successfully write/read any specific cell in the array? If yes, will using an access device affect the integration density in the array scenario?”

Reply-3:

Yes. As mentioned in the introduction, there are many potential hazards on the way from a single cell to RAM (as can be learned from the history of MRAM). Some sort of isolation would be needed for avoiding sneak paths. We have two possible strategies in mind. The first approach is based on physical isolation, using electrically isolated control lines (CL) in a different layer. A similar strategy is used in the existing RSFQ RAM. The CL's create magnetic fields that enable cell selection both for write and read operations. Implementation of CL's does not strongly affect the footprint, but requires vertical stacking of layers. The

second approach is based on logical (hardware) isolation. E.g., in MRAM the selection is achieved by implementation of an additional isolating transistor for each cell. Unfortunately, there is no superconducting transistor with good-enough isolation. However, recently we have demonstrated reconfigurable vortex-based superconducting diodes with high nonreciprocity [32]. Potentially such programmable diodes can be employed for avoiding sneak paths in a RAM mesh. However, this comes at an expense of inevitable growth of complexity and footprint. At present, the second strategy looks more cumbersome. Nevertheless, diodes can be very instrumental for multiplexing. A clarification is added in the new sec. of the Supplementary: “VIII. Possible strategies for building a multibit AVRAM”, with a reference in the concluding part of the paper.

Reviewer #2 writes:

“- It is not clear how the write/read operation of a specific cell in an array can be done without disturbing other cells. How will the WL and BL pulses be applied in write and read operations? What will be the sensing mechanism?”

Reply-4:

The answer to the first part of the question is provided in Reply-3. The selection for readout could be made with the help of control lines. By sending a small current via the N-th CL it is possible to slightly reduce critical currents to make $I_c^*(state-1) < I_c(state-1)$, but keep $I_c^*(state-0) > I_c(state-1)$ [here *-indicates reduced values in the presence of CL currents]. In this case, the probe current through the M-th BL, $I_c^*(state-1) < I < I_c(state-1)$ will activate only the NxM cell. If the readout voltage is 0/finite than the NxM cell is in state 0/1. The corresponding margins are fairly high due to the large difference of $I_c(0)$ and $I_c(1)$ in our cells. For example, the $L_x=3$ μm cell from Fig. 1(a) has $I_c(0)=250$ μA and $I_c(1)=120$ μA . If CL creates the offset field of -1 Oe, $I_c^*(0)=200$ μA and $I_c^*(1)=75$ μA , as can be seen from $I_c(H)$ from Figs.1 (d) and (e),. Therefore, the probe current has a significant tolerance, 97.5 ± 22.5 μA . The margins can be further broadened by proper geometrical design. A clarification is added in sec. VIII of the Supplementary.

Reviewer #2 writes:

“- The authors only show the write operation of the memory cell. They should also show the read operation.”

Reply-5:

The readout procedure was partly explained in the Supplementary (Sup.Fig.4). to meet the Reviewer’s critics, we added additional discussion in line with replies 3 and 4 above.

Reviewer #2 writes:

“- The manuscript does not provide enough discussion on the existing cryogenic memory technologies and their limitations.”

Reply-6:

It is almost impossible to make a comprehensive and fair overview within a short available space of the introduction. Instead, we refer to a recent comprehensive review on cryogenic memory technologies [29] (the reference is updated). So far, many of proposed new technologies exist only as ideas and are waiting for the experimental scrutiny. Therefore, it is too early to talk about their limitations. We don’t want to discourage researchers from trying: at this stage it is important to try many and different approaches.

Reviewer #3:

Reviewer #3 writes:

“The authors experimentally demonstrated the operation of submicron superconducting memory cells based on an Abrikosov vortex (AV) trapped in the superconducting film. They placed planer-type Josephson junctions close to a trap center and observed the change of a Josephson threshold current with and without AV in the trap center. Three significant results were shown in the article. Firstly, they showed the stable write/read operation in the memory cell of 1 μm size. Second, they demonstrated controllable write/read operation in the memory cell by introducing the asymmetrical cell structure. Finally, they showed the selectivity of the memory cell through bit-line and word-line, where applying the bit-line current to the cell considerably reduced the word-line current for data writing. These features are promising for fabricating high-dense superconducting memories that are not realized yet. However, several questions and comments shown below about the proposed system and experimental results have to be clarified.

1. The reason for the large size of the RSFQ circuit is due to the inductance formed on the ground plane. This is effective for reducing magnetic coupling between memory cells. In a memory using vortices on a plane, it is considered that the effect of magnetic coupling between memory cells affects the memory operation as the memory scale increases. The authors need to comment on how to solve this.”

Reply-1:

Magnetic crosstalk via stray fields indeed occurs. However, we don't think that this is critical for the ultimate miniaturization for two reasons: 1) Stray fields are caused by the large demagnetization factor in the planar geometry. The spatial extent of stray field is determined by the thickness of the film, which corresponds to the effective length of magnetic dipole. In Nb films it can be made very small, in the 10 nm range. Therefore, stray fields will rapidly decay at the ~ 100 nm distance, which is the estimated limit for the miniaturization. 2) In the mesoscopic case, when the size of the island becomes smaller than the Pearl length, ~ 500 nm for our films, the flux in the vortex is no longer quantized. The net flux decreases proportionally to the square of the ratio of the island size to the Pearl length. This will have a corresponding effect on further reduction of stray fields, which disappear in the deep mesoscopic limit when the vortex does not carry flux at all, thus eliminating stray fields. We added a short note in the text and a longer discussion in the Supplementary (the end of sec. VIII).

Reviewer #3 writes:

“2. The R versus I_{WL} characteristic in Fig. 3(b) becomes oscillatory when I_{WL} exceeds the threshold. Also, the period of the oscillations is irregular. What causes this oscillation?”

Reply-2:

A possible origin of stratification is discussed in the Reply-1 to Reviewer-1: It is likely the stroboscopic effect due to a relatively long pulse duration. We added a corresponding clarification in the modified version of the Supplementary (sec. VII. Stratification of switching diagrams) and a short note in the main text.

Reviewer #3 writes:

“3. Why does applying bit-line current I_{BL} reduce the total threshold current I_{tot} in the word and bit-line operation? A qualitative explanation is necessary.

4. When the bit-line current I_{BL} was zero, the word-line currents required for the writing “-1” operation and the data-erasing operation were $I_{WL} = -730\mu\text{A}$ and $150\mu\text{A}$, respectively, as shown in Table 1. On the other hand, when $I_{BL} = -300\mu\text{A}$, $I_{WL} = 29\mu\text{A}$ and $200\mu\text{A}$, respectively. Why did the application of I_{BL} reverse the sign of the word-line current for the writing “-1” operation? Also, why did the current for the data-erasing operation increase the word-line current? A qualitative explanation is necessary.”

Reply-3,4:

Q3 and the part of Q4 are related; therefore, we provide a combined answer. The sign-reversal of I_{BL} is a very important observation, which gives a clue to understanding of the cooperative WL+BL effect. We are grateful to the Reviewer for pointing this out. As shown in Fig. 5(c), 0-> -1 writing can be made by simultaneous $I_{BL} = -300\mu\text{A}$ and a very small POSITIVE $I_{WL} = 29\mu\text{A}$. A logical explanation is that the finite I_{BL} opens a new easy channel for vortex entrance: not along the track from the bottom of the cell, but from the opposite side, i.e., from the top readout junction. Most likely, this involves a Josephson vortex, that penetrates into the readout JJ and then pushed into the trap. The positive I_{WL} creates a negative H_y field component at the readout JJ. The negative I_{BL} creates a positive/negative H_y at the left/right edges of the readout JJ. Therefore, negative I_{BL} + positive I_{WL} will generate a larger negative H_y at the right edge of the readout JJ. Thus, a Josephson antivortex will enter the JJ from the right edge. Subsequently, it will be pushed to the center of the JJ by the negative I_{BL} and from there make a small jump down to the trap with the help of the positive I_{WL} . This scenario, in combination with some additional left-right asymmetry (see the Reply-5), is consistent with the observed WL+BL operation, reported in Fig. 5(c) and Suppl. Fig. 5. A clarification is added in the modified version of the Supplementary (sec. V) + a short note in the main text.

Concerning the last question, Table-I lists threshold currents above which operations can be performed. For example, for $I_{BL}=0$, 0 -> -1 switching requires $I_{WL} < -730\text{ uA}$ and -1 -> 0 occurs at $I_{WL} > 150\text{ uA}$. The erasing -1 operation at $I_{WL}=200\text{ uA}$ and $I_{BL}=0$, demonstrated in Fig. 6, does not mean that the threshold current has changed, but simply means that 200 uA is above the threshold (150uA) and, therefore, does this operation. The subsequent 200uA pulses do not do anything because there is nothing to erase. A note is added in the modified version.

Reviewer #3 writes:

“5. Why are the magnitudes of the threshold word-line current different for the positive and negative bit-line current I_{BL} in Fig. 5(c)?”

Reply-5:

Good point. As shown in the Suppl. Fig.5, the positive I_{WL} leads to 0-> -1 switching for both polarities of I_{BL} . Thus, in both cases an antivortex is pushed into the trap from the top readout JJ. This supports the scenario discussed in Reply-4: the positive I_{WL} pushes the Josephson antivortex downwards and, therefore, leads to 0-> -1 switching. However, the observed asymmetry of threshold I_{WL} currents for positive and negative I_{BL} must indicate the presence of an additional left-right asymmetry in the readout JJ, due to which it is easier for Josephson vortices to enter from the right side of the readout JJ. Some left/right asymmetry can indeed be seen in the SEM image Fig. 2(a). This again indicates the importance of conscious geometrical design of such mesoscopic fluxonic quantum dots. A clarification is added in the modified version of the Manuscript and the Supplementary.

Reviewer #3 writes:

“6. In Fig. 6 (b), if the authors examine the robustness of the memory operation, the threshold for the data-erasing operation concerning the application of positive I_{WL} should be examined.”

Reply-6:

The corresponding data is shown in the Supplementary Fig. 6 (b). Here -1 \rightarrow 0 erasing operation is performed by $I_{WL} = +200$ μ A and 1 \rightarrow 0 erasing by $I_{WL} = -200$ μ A. The -1/1 states can be distinguished from readout JJ resistances, which is slightly higher for 1 state.

REVIEWER COMMENTS

Reviewer #1 (Remarks to the Author):

I have checked the author's responses and the corresponding corrections. I have no further questions and recommend the publication.

Reviewer #2 (Remarks to the Author):

- The authors mention that in their case, the scenario of flux trapping will be the opposite compared to the other superconducting devices. However, they missed my query about inductive coupling. Will their device be sensitive to inductive coupling? If yes, while building large arrays, will keeping a finite distance between multiple cells be required to avoid unintentional write and read of the unaccessed cells?

- The authors mention that the reason behind the two different critical current levels in Fig. 1(c) and Fig. 3(a) is the different junction lengths. However, the insets of both figures show the same length (1 μm).

- The authors have superficially responded to my query about the array scenario. They should show a specific array design for their memory device and clearly mention how a given cell will be written and read from the array. How will the WL and BL biasing be chosen? They mentioned two possible solutions but did not provide sufficient technical discussion. They claim to show WL and BL operations in the title of the manuscript without testing/simulation the array-level dynamics.

- The authors mention that using a control line is feasible in the array scenario. However, they never mentioned how they will connect and use the control line to write and read any specific cell in the array.

- In the added supplementary discussion, the use of the word 'multibit' is misleading and technically incorrect. Typically, a memory cell is referred to as 'multibit' if one cell in the array can alone store more than one bit of information which is not the case here.

- Finally, the authors avoided my query about the sensing mechanism of this memory. How do the authors propose to sense the memory state stored in a specific cell in the array during read operation?

Reviewer #3 (Remarks to the Author):

The authors have provided appropriate answers to all of the reviewer's questions, and I believe this paper is acceptable for publication.

2nd reply to the Reviewer #2

1. Reviewer #2 writes:

”- The authors mention that in their case, the scenario of flux trapping will be the opposite compared to the other superconducting devices. However, they missed my query about inductive coupling. Will their device be sensitive to inductive coupling? If yes, while building large arrays, will keeping a finite distance between multiple cells be required to avoid unintentional write and read of the unaccessed cells?”

Reply 1.

We apologize for the confusion: the answer to this question was provided in the end of sec. VIII of the Supplementary, but was not articulated in the Reply. In the modified version we expanded this discussion and put it in a dedicated section IX of the Supplementary:

IX. Crosstalk and AVRAM density

Apart from the size of a single cell, the AVRAM density can be also affected by the crosstalk between neighbor cells. The crosstalk can be caused either by vortex stray fields or by inductive coupling between electrodes.

We believe that the crosstalk via vortex stray fields should not be critical for ultimate miniaturization of AVRAM for two reasons. Firstly, stray fields are caused by the large demagnetization factor in planar geometry and their spatial extent is determined by the thickness of the film, which corresponds to the effective length of the magnetic dipole. It can be made very small, in the 10 nm range, greatly reducing stray fields at the estimated limit of miniaturization of about 100 nm. Secondly, in the mesoscopic case, when the size of the electrode becomes comparable or smaller than the Pearl length ($\lambda_P \approx 500$ nm for our films [34]), the flux in the vortex is no longer quantized. The net flux decreases proportionally to the square of the ratio of the island size to the Pearl length, resulting in a corresponding reduction of stray fields. In the deep mesoscopic limit, $L_{x,z} \ll \lambda_P$, the vortex does not carry any flux at all, thus eliminating vortex stray fields.

There could also be an inductive crosstalk via control lines: the control-line current will induce some field at nearby cell rows as well. This type of crosstalk can be mitigated in two ways: 1) Improving the half-selection stability: if it is good enough the modest crosstalk field would not be critical. As demonstrated earlier [7], the half-selection stability of AVRAM cells can be very good. The observed cooperative WL+BL effect further improves the stability. 2) Proper geometrical design. Here the point is that planar structures respond only to the perpendicular component of magnetic field. The control line generates fairly uniform parallel fields above and below the CL, but highly nonuniform (almost singular) perpendicular field component at the edges, which decay at the scale of the film thickness. Therefore, the crosstalk via perpendicular field component can be made very short-range (~ 10 nm) with the corresponding minimal influence on the RAM density.

2. Reviewer #2 writes:

”- The authors mention that the reason behind the two different critical current levels in Fig. 1(c) and Fig. 3(a) is the different junction lengths. However, the insets of both figures show the same length (1 μ m).”

Reply 2:

The critical current of our variable thickness-type junctions is proportional to the length and inversely proportional to the remaining thickness of the constricted part of the film, which is estimated to be ~ 15 -20 nm. The latter is the major factor in determination of I_c because a relatively small decrease of the thickness could lead to a significant reduction of I_c . The main difference of the devices from Figs. 1(b) and 2(a) is that in the one from Fig. 2(a) the excessive area of Nb film has been removed by additional FIB milling. The associated extra ion etching inevitably affected the junctions, which were unintentionally etched during multiple FIB scans needed for alignment of the etch patterns. The corresponding noticeable reduction of I_c for the device from Fig. 2 (a) indicates that the remaining thickness of the constriction in this junction is ~ 6 -8 nm thinner than for the one in Fig. 1 (b). A clarification is added in Methods.

3. Reviewer#2 writes

”- The authors have superficially responded to my query about the array scenario. They should show a specific array design for their memory device and clearly mention how a given cell will be written and read from the array. How will the WL and BL biasing be chosen? They mentioned two possible solutions but did not provide sufficient technical discussion. They claim to show WL and BL operations in the title of the manuscript without testing/simulation the array-level dynamics.”

4. ”- The authors mention that using a control line is feasible in the array scenario. However, they never mentioned how they will connect and use the control line to write and read any specific cell in the array.”

5. ”- Finally, the authors avoided my query about the sensing mechanism of this memory. How do the authors propose to sense the memory state stored in a specific cell in the array during read operation?”

Reply 3-5:

Questions 4-6 are related because they concern operation of RAM arrays. Therefore, we provide a combined answer. As we mentioned in the previous version, RAM cells would require isolation between WL (CL) and BL in order to avoid current sneak paths. We claimed that such RAM would allow individual addressing of $N \times M$ cell by sending proper currents through the N -th control line and M -th bit line, both for write and read operations. In order to address the issue more carefully, we made new cells with isolated CL and BL. The subsequent experimental analysis and extended discussion are added in the significantly expanded sec. VIII of the Supplementary. They include two new Figures, S8 and S9 and confirm/clarify our statements. Here is the full text of the modified section VIII with answers to Q3-5:

VIII. Strategy for building AVRAM arrays

There are many potential hazards on the way from a single cell to a multi-cell RAM. The problems are caused by mutual crosstalk between cells, either via current sneak paths in a RAM array or via inductive coupling of cells and electrodes. To avoid this, some sort of isolation between WL and BL is needed. We have two possible strategies in mind. The first is based on a physical isolation with implementation of additional, electrically isolated control lines (CL's), as sketched in Supplementary Figure 8 (a). Such a strategy is used in the existing RSFQ RAM.

Supplementary Figure 8. **Operation of AVRAM with electrically isolated control and bit lines.** (a) A sketch of AVRAM arrays with electrically isolated CL and BL. The $N \times M$ cell selection for both write and read operation is achieved by coincident current pulses through N -th CL and M -th BL. (b) SEM image (false color) of a cell with electrically isolated control line and bit line. (c) Demonstration of $0 \rightarrow 1$ write operation by CL and BL current pulses for the cell from (b). Here the read out junction resistance is shown as a function of the CL pulse amplitude for a fixed BL pulse amplitude ~ 1 mA. It is seen that the general behavior is similar to the WL + BL operation, discussed in the paper.

To test this idea, we fabricated cells with electrically isolated WL(CL) and BL. Supplementary Fig. 8 (b) shows a SEM image of such the cell. It has a three-layer structure with two Nb electrodes separated by an insulating SiO_2 layer. The bottom (horizontal) Nb electrode forms a CL and a top (vertical) - the BL, which contains the same type of memory cell as discussed in the paper. We observe that the cells with isolated CL and BL behave very similar to the cells with interconnected WL and BL, discussed in the paper. The CL creates magnetic fields that enable cell selection for both write and read operations.

Supplementary Fig. 8 (c) demonstrates the $0 \rightarrow 1$ write operation with coincident CL and BL current pulses for the cell from Fig. S8 (b). Here the readout junction resistance is shown as a function of the CL pulse amplitude at a fixed BL current pulse amplitude ~ 1 mA. It is seen that the general behavior is similar to the WL+BL operation, discussed in the paper. The threshold CL current ~ 2 mA is also similar to the WL current for the same-size cells, see Fig. S7 (b) and Ref. [7]. This indicates that the primary physical effect of WL/CL currents is the generation of magnetic field, which does not depend on the interconnection with the BL (this is the well-known short-circuit principle in multilayer superconducting electronics). Therefore, all conclusions made in the paper are fully applicable to cells with isolated CL's.

Since there are no sneak paths in AVRAM arrays with isolated CL's and BL's, cell selection for write operation is straightforward, as sketched in Fig. S8 (a). The remaining issue is the selection for readout. Here the problem is that the probe current is sent through all cells on the specific BL and the readout voltage is measured over all JJs on the BL, connected in series. Therefore, selective readout of a specific cell requires that all other cells on this BL remain in the zero-voltage state. Consequently, the probe BL current should be smaller than critical currents, $I_c(0, 1)$, in both 0 and 1 states.

Supplementary Figure 9 explains the procedure for selective readout for RAM arrays as in Fig. S8 (a). Left panel shows sketches of the I - V 's of readout JJ in 0 and 1 states in the absence of CL current. Parameters correspond to the $L_x = 3 \mu\text{m}$ cell from Fig. 1(a), which have $I_c(0) = 250 \mu\text{A}$ and $I_c(1) = 120 \mu\text{A}$. The readout selection can be achieved with the help of CL. By sending a small current via the N -th CL, it is possible to reduce critical currents so that $I_c^*(\text{state-1}) < I_c(1)$, but $I_c^*(\text{state-0}) > I_c(1)$ [here * indicates reduced values in the presence of CL currents]. The corresponding I - V 's are drawn in the right panel of Fig. S9. They correspond to a small offset field $H_{\text{CL}} = -1$ Oe, at which $I_c^*(0) = 200 \mu\text{A}$ and $I_c^*(1) = 75 \mu\text{A}$, as can be seen from the $I_c(H)$ patterns in Figs.1 (d) and (e). The green band indicates margins of the readout current through the M -th BL, $I_c^*(1) < I_{\text{BL}} < I_c(1)$, which will activate only the $N \times M$ cell: if the readout voltage is 0/finite, then this cell is in state 0/1. All other cells will remain in the zero-voltage state because $I_{\text{BL}} < I_c(0, 1)$. The corresponding margins are fairly broad due to the

large difference of $I_c(0)$ and $I_c(1)$ in our cells. For the considered example of $H_{CL} = -1$ Oe, the probe current has significant tolerance margins, $I = 97.5 \pm 22.5 \mu\text{A}$, which can be further widened by proper geometrical design. Thus, RAM arrays with isolated control and bit-lines is the simplest design which facilitates selective addressing of the cells.

Supplementary Figure 9. **Clarification of the selective readout procedure.** Left panel shows the current-voltage characteristics in 0 and 1 states in the absence of control line current. Right panel shows the same I - V 's with a small current through the N -th CL, creating an offset field, $H_{CL} = -1$ Oe. The green band indicates the margins of the readout current $I_c^*(1) < I_{BL} < I_c(1)$ through the M -th BL, which will induce the finite readout voltage only if the $N \times M$ cell is in the 1-state. Parameters in this example correspond to the $L_x = 3 \mu\text{m}$ cell from the main manuscript.

The implementation of CL's does not strongly affect the cell footprint but requires vertical stacking of layers. The alternative approach for building RAM is based on logical (hardware) isolation. For example, in MRAM, the selection is achieved by adding a switching transistor for each cell. Unfortunately, there is no superconducting transistor with good enough isolation. However, recently we have demonstrated reconfigurable vortex-based superconducting diodes with high nonreciprocity [32]. Potentially, such programmable diodes can be employed to reduce sneak paths in a RAM bank. However, this comes at an expense of inevitable growth of complexity and footprint. At present, the second strategy looks more cumbersome. Nevertheless, diodes can be very instrumental for multiplexing.

6. Reviewer#2 writes:

“- In the added supplementary discussion, the use of the word ‘multibit’ is misleading and technically incorrect. Typically, a memory cell is referred to as ‘multibit’ if one cell in the array can alone store more than one bit of information which is not the case here.”

Reply-6:
Thank you, corrected.

Finally, we want to thank the Reviewer for valuable remarks. In reply to Reviewers questions we made extensive changes, fabricated new sample, made new measurements, provide additional data and clarifications. Since the other two Reviewers were satisfied with the previous version of the manuscript, we have chosen not to introduce major changes to the manuscript. Therefore, main changes and clarifications are added to the Supplementary information. The changes are red-marked in the resubmitted files.

REVIEWERS' COMMENTS

Reviewer #2 (Remarks to the Author):

Thanks for addressing my concerns.

Reply to the Reviewer:

Thank you, your comments encourage us to proceed with our work.